# The tomato receptor CuRe1 senses a cell wall protein to identify *Cuscuta* as a pathogen

Volker Hegenauer [1,2], Peter Slaby[1], Max Körner[1], Julien-Alexander Bruckmüller [3,9], Ronja Burggraf[2], Isabell Albert[1], Bettina Kaiser[2], Birgit Löffelhardt[2], Irina Droste-Borel[4], Jan Sklenar [5], Frank L. H. Menke [5], Boris Maček [4], Aashish Ranjan [6,7], Neelima Sinha [6], Thorsten Nürnberger[2,8], Georg Felix[2], Kirsten Krause [3], Mark Stahl[2] & Markus Albert [1✉]

Parasitic plants of the genus *Cuscuta* penetrate shoots of host plants with haustoria and build a connection to the host vasculature to exhaust water, solutes and carbohydrates. Such infections usually stay unrecognized by the host and lead to harmful host plant damage. Here, we show a molecular mechanism of how plants can sense parasitic *Cuscuta*. We isolated an 11 kDa protein of the parasite cell wall and identified it as a glycine-rich protein (GRP). This GRP, as well as its minimal peptide epitope Crip21, serve as a pathogen-associated molecular pattern and specifically bind and activate a membrane-bound immune receptor of tomato, the Cuscuta Receptor 1 (CuRe1), leading to defense responses in resistant hosts. These findings provide the initial steps to understand the resistance mechanisms against parasitic plants and further offer great potential for protecting crops by engineering resistance against parasitic plants.

[1] Department of Biology, Molecular Plant Physiology, Staudtstraße 5, 91058 Erlangen, Germany. [2] Center for Plant Molecular Biology, Auf der Morgenstelle 32, 72076 Tübingen, Germany. [3] Department of Arctic and Marine Biology, Breivika, 9037 Tromsø, Norway. [4] Quantitative Proteomics & Proteome Center Tübingen, Auf der Morgenstelle 15, 72076 Tübingen, Germany. [5] The Sainsbury Laboratory, University of East Anglia, Colney Lane, NR4 7UH Norwich, UK. [6] Department of Plant Biology, College of Biological Sciences, UC Davis, Davis, CA, USA. [7] National Institute of Plant Genome Research, New Delhi, India. [8] Department of Biochemistry, University of Johannesburg, Johannesburg 2001, South Africa. [9] Present address: Solana Research GmbH, Eichenallee 9, D - 24340 Windeby, Germany. ✉email: markus.albert@fau.de

C haracteristic molecular patterns uncover pathogens as external invaders and are critical signatures that are detected by the innate immune system of both, animals and plants. This discrimination between self and non-self allows the host organisms to initiate defense reactions and resist pathogen attacks. As part of their innate immune system, plants evolved cell surface receptors to detect molecular patterns[1–3]. Due to the facts that most plant pathogens are microbes or arthropods and the wider evolutionary distance between plants and those attackers, the presence of "plant pattern recognition receptors" to detect molecular patterns seems a logical consequence of evolution. However, ~4500 higher plant species live as parasites and thus pose an additional threat to plants. Well-known parasitic plants with high agronomical relevance are *Striga* spp., *Orobanche* spp., and *Cuscuta* spp[4–6]. Most host plants are unable to detect an invasion by parasitic plants and the attackers stay unrecognized due to the limited innate immune system of host plants for detecting dangerous parasitic plants. Yet, a few host exceptions are described, which are able to fend off parasitic plants and stay incompatible[7–9]. However, the molecular mechanisms behind these are poorly understood, and molecular patterns of parasitic plants, which could mark a plant parasite as a devastating invader, have not yet been described.

*Cuscuta* spp. are holoparasitic plants which infect a broad spectrum of hosts by connecting to their vasculature via specific feeding structures, called haustoria (Fig. 1a, b)[10–12]. Cultivated tomato (*Solanum lycopersicum*) is one of few host exceptions that recognizes *Cuscuta reflexa* as an alien invader and actively initiates defense responses[4,13,14] measureable as the emission of the stress-phytohormone ethylene or reactive oxygen species (ROS) in tomato leaves[15] and visible as hypersensitive response (HR) at the infection sites (Fig. 1c)[13–15]. We reported the tomato cell surface receptor "Cuscuta receptor 1" (CuRe1) as a critical component for the detection of *C. reflexa* due to a hypothesized Cuscuta factor or pathogen-associated molecular pattern (PAMP) that can be extracted from the parasitic plant and triggers the defense response in a CuRe1-dependent manner. This Cuscuta factor is a heat stable protein and sensitive to treatments with bases (pH > 11), indicating potential secondary modifications present on the peptide backbone[16]. The Cuscuta factor is found in all organs of *C. reflexa* irrespective of its infectious stage and seems to locate to the parasite's cell wall[15]. Here, we purified this Cuscuta factor from *C. reflexa* extracts and identified it as a Glycine-rich protein (GRP). We further characterized its function as a binding ligand for CuRe1 and the specifically triggered plant defense responses.

## Results and discussion
### Purification and identification of a defense-triggering Cuscuta factor.
Since we knew that the Cuscuta factor originates from the cell wall, we focused on extracts prepared from the parasite cell wall and tested them for bioactivity in the ethylene bioassay specifically induced via CuRe1 (Fig. 1d). Compared to incubation in buffer/water alone, higher amounts of Cuscuta factor were found to be released from cell wall fractions by treatments with pectinases and, with much lower efficiency, by cellulases (Fig. 1d). Both types of enzymes are known to be present and active in penetrating *Cuscuta* spp. haustoria and can thus lead to an increased release of the Cuscuta factor from the cell walls of intruding haustoria during the infection process[17,18].

To extract sufficient amounts of the Cuscuta factor from collected plant material, we scaled up the previous protocol[15] and used acidic extraction conditions (0.1 M HCl, pH 1). The analysis was also extended to all of the activities that eluted as distinct peaks from the first cation exchange column (Supplementary Fig. 1).

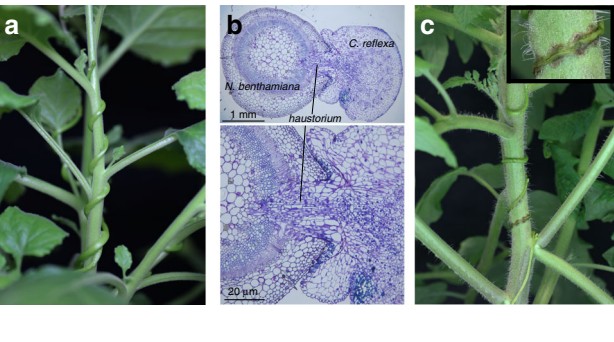

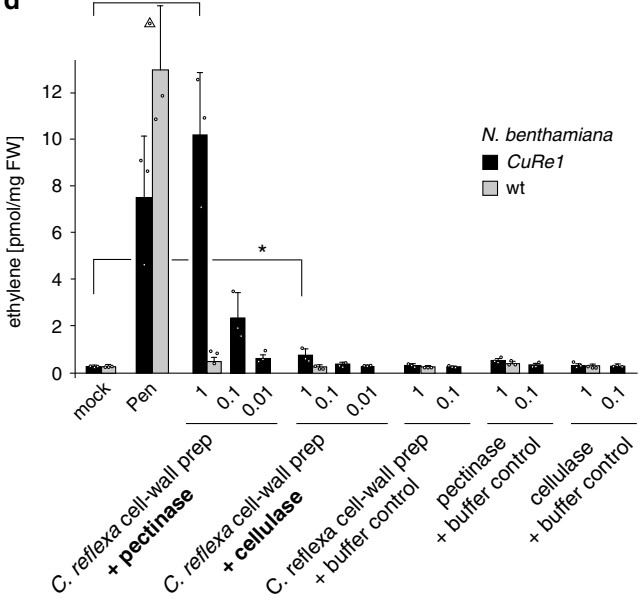

**Fig. 1 *Cuscuta reflexa* induces defense in cultivated tomato by a pathogen-associated molecular pattern. a** *C. reflexa* can infect nearly all dicot plants as susceptible hosts; picture shows infected *Nicotiana benthamiana*; **b** Parasitic haustoria successfully penetrate the *N. benthamiana* shoot; **c** *C. reflexa* induces visible hypersensitive response (HR) on tomato (*S. lycopersicum*) shoot at the contact sites of the parasite's haustoria. **d** *C. reflexa* cell wall preparations were treated with either cellulase or pectinase; extracts were applied to trigger defense-related ethylene biosynthesis in transgenic CuRe1-expressing *N. benthamiana* plants. Numbers on x-axis indicate applied extract volume in µl; bovine serum albumin (BSA; 0.01 mg/ml) buffered in 25 mM MES (pH 5.7) was added as mock control; *Penicillium* extract (0.05 mg/ml) served as positive control[31]. FW, fresh weight. Ethylene measurements show means of three technical replicates; error bars denote SD. Wildtype (wt) *N. benthamiana* plant samples have not been tested with diluted extract preparations (0.1 µl and 0.01 µl) since they did not respond to maximum doses (1 µl) in the ethylene assay. Dots indicate single data points, triangle shows outlier at 16.87; Asterisks show Student's *t* test; ***$p < 0.0028$; *$p < 0.0285$; representative graphs are shown; all experiments were repeated more than three times.

When purifying the extracts by cation exchange or reversed phase chromatography, the Cuscuta factor activity detectable by the CuRe1 receptor eluted in several peaks, indicating presence of activity in structurally different forms (Supplementary Fig. 1)[15]. We further purified and enriched the Cuscuta factor(s) from the obtained fractions and performed LC-MS/MS analyses for each sample individually. Several distinct masses correlated with CuRe1-dependent bioactivity and we identified 11 different compounds all of which represented active forms of the Cuscuta factor (Supplementary Fig. 2). MS/MS fragmentation studies of the correlated candidate masses shared similar fragment peaks

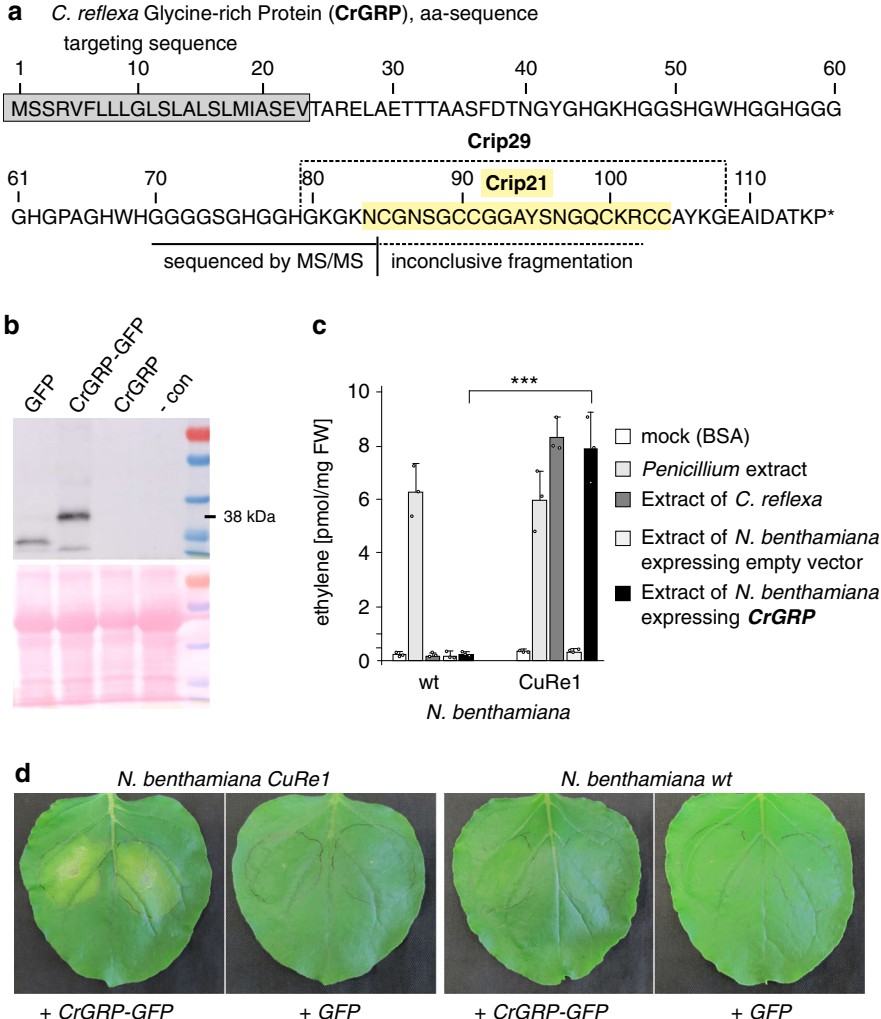

**Fig. 2 The *C. reflexa* Glycine-rich protein (CrGRP) triggers CuRe1-dependent defense responses. a** Protein sequence of the identified CrGRP; the peptide isolated from *C. reflexa* extracts was sequenced de novo by mass spectrometry and is indicated starting at aa-position 70, dashed line indicates the peptide part which could not be sequenced by MS/MS; Crip21 motif highlighted in yellow. **b** heterologous expression of *CrGRP* gene in *N. benthamiana* leaves; WB shows a c-terminally GFP-tagged CrGRP. **c** Ethylene response in leaves of wt or *CuRe1*-expressing *N. benthamiana*. Plants were treated with extracted CrGRP after heterologous expression shown in b. Bovine serum albumine (BSA; 0.01 mg/ml) buffered in 25 mM MES (pH 5.7) was added as mock control; *Penicillium* extract (0.05 mg/ml) served as positive control[31]. FW, fresh weight; ethylene measurements show means of three technical replicates; dots indicate single data points; error bars denote SD, asterisks denote student's t-test, p ≤ 0.0003; representative graphs are shown; experiments were repeated more than three times. **d** Expression of CrGRP-GFP in leaves of wildtype (wt) and transgenic *CuRe1*-expressing *N. benthamiana* plants. GFP alone served as negative control; pictures 5 days after expression.

(Supplementary Fig. 2) and had the fragment mass of 2077 Da in common. Another characteristic feature of all corresponding MS-spectra was an inconclusive fragmentation pattern with only a few clear but characteristic fragment masses (Supplementary Fig. 2) that have been previously observed[15]. Since the commonly present fragment of 2077 Da is a y-fragment and the spectra of the heavier candidate masses contained the lighter candidate masses as their y-fragments, we assumed a common origin of all fragments from the same protein. The mass differences of the N-terminal fragmentations could be correlated to those of single amino acid residues, which allowed us to deduce the N-terminal part of the peptide sequence (Supplementary Fig. 2). The information obtained from overlays of seven individual MS/MS fragmentation analyses in four individual LC-MS/MS runs revealed the sequence of the first 15 N-terminal amino acids of the peptide (Fig. 2a and Supplementary Fig. 2). A p-blast search against a translated transcriptome database from *C. reflexa*[19] resulted in a perfect hit on a glycine-rich protein (GRP) of *C. reflexa* (Fig. 2a). CrGRP consists of 116 amino acids

with an n-terminal targeting sequence that predicts an extracellular localization (Fig. 2a). According to the current classification of GRPs[20], the CrGRP belongs to the class II which comprises a distinguishing c-terminal cysteine-rich region (Fig. 2a). We cloned the corresponding *CrGRP* gene from *C. reflexa* genomic DNA and transiently expressed it in *N. benthamiana* leaves for ~72 h with c-terminal GFP or tagRFP fusion tags. We confirmed the predicted localization of GRP within the cell wall with confocal microscopy of *N. benthamiana* leaves transiently expressing a tagRFP-tagged version of the CrGRP (Supplementary Fig. 3). Western blot analyses showed that the protein migrated at the calculated size and does not appear to be secondarily modified in *N. benthamiana* (Fig. 2b). Extracts of these leaves, expressing *CrGRP*, induced ethylene production in a CuRe1-dependent manner like the original *C. reflexa* extract (Fig. 2c). Moreover, when expressing *CrGRP* in *N. benthamiana* leaves for 5–7 days, clear hypersensitive cell death can be observed only when CuRe1 is present but not in control plants (Fig. 2d) lacking the receptor. These findings demonstrate that

CrGRP is the trigger to initiate the CuRe1-dependent defense program.

**The minimal peptide epitope of CrGRP**. Due to unspecific degradation of the full-length CrGRP during our initial extraction protocol, the extracted forms of the Cuscuta factor were rather small peptides in a range between 2000 and 4000 Da[15] (Supplementary Table 1). We therefore assumed a minimal peptide motif within the CrGRP full-length protein (~11.5 kDa), which must be sufficient to trigger the defense program. We thus tested a synthetic peptide representing CrGRP$_{82-106}$ for activity. The peptide was highly active and triggered ethylene production at concentrations ≥0.1 nM (Fig. 3a, b), which is in the range we previously estimated for the Cuscuta factor purified from plant extracts[15]. To further narrow down the active motif we tested shortened versions of the peptide, resulting in a 21 aa peptide, termed as crip21 for cysteine-rich peptide 21, that still retained the full activity found with CrGRP$_{82-106}$ or full-length CrGRP (Figs. 2a and 3a, b). Peptides further shortened from the N- or C-terminus showed only reduced activity or no activity, respectively (Fig. 3a and Supplementary Fig. 4). A 15 aa peptide representing the N-terminal sequence present on the purified peptide was inactive (Supplementary Fig. 4), further demonstrating that activity resides in the C-terminal part with the Crip21 motif of CrGRP (Fig. 2a).

Treatment of *C. reflexa* extracts or purified Cuscuta factor with NH$_4$OH at pH ≥ 11 leads to a total loss of bioactivity and indicated potential secondary modifications on the CrGRP[15,16]. However, synthesized peptides trigger responses in a picomolar range (Fig. 3b and Supplementary Fig. 4) suggesting that no such secondary modifications are required for the bioactivity of Crip21. Much like the purified Cuscuta factor[15], synthesized Crip21 peptide, comprising no secondary modifications, lost all of its activity when treated with NH$_4$OH (Supplementary Fig. 5a). We analyzed the NH$_4$OH-treated peptide by MS/MS and observed multiple reaction products of Crip21 among which the most prominent ones were 82 or 99 Da smaller (Supplementary Fig. 5b), clearly showing that the treatment of Crip21 with NH$_4$OH is modifying the peptide itself.

We infiltrated Crip21 into the leaves of resistant *S. lycopersicum*, susceptible *Solanum pennellii* and the introgression line IL8-1-1 lacking *CuRe1*[15], to check whether Crip21 also induces visible HR in tomato. After 7 days, only the cultivated tomato or an introgression line (IL) with functional CuRe1 showed Crip21-dependent cell death while *S. pennellii* and the IL lacking CuRe1 did not (Fig. 3c).

**Binding of CrGRP and Crip to the receptor CuRe1**. To test for a direct interaction of the peptide epitope Crip with the receptor CuRe1, we n-terminally labelled a 29-aa-long Crip peptide with biotin (bio-Crip29). The bio-Crip29 peptide is an N-terminally (+4 aa) and C-terminally (+4 aa) prolonged Crip21 (Fig. 2a) and was as active as Crip21 (Fig. 3b). The Crip21 minimal epitope has been prolonged to introduce a higher number of Lysine residues to increase the chance for a successful chemical crosslinking of NH$_2$ groups on the ligand with NH$_2$ groups on the receptor. We then examined the interaction of CuRe1 proteins with bio-Crip29 in affinity-crosslinking experiments *in planta*. *N. benthamiana* leaves expressing the myc-tagged receptor CuRe1 were first incubated with the bio-Crip29 derivative, either alone or together with an excess of non-modified Crip as competitor, and the leaves were subsequently treated with a chemical cross-linker. When analyzed for the presence of biotin, immunoprecipitates of CuRe1 showed clear labelling which was absent in samples treated with an excess of non-modified Crip as competitor (Fig. 3d). In turn, binding of bio-Crip29 was not out-competed when using structurally unrelated

peptides such as flg22 (Fig. 3d). These findings demonstrate the CrGRP derived peptide epitope Crip as the specific ligand for the CuRe1 receptor. To corroborate direct protein–protein interaction of the full-length CrGRP with CuRe1 as it may occur under physiological conditions, both proteins were co-expressed with different c-terminal tags and the interaction of both could be demonstrated in co-immunoprecipitation assays (Fig. 3e).

To identify aa-residues of Crip21 which are critical for CuRe1-activation, the 21 aa-residues were individually substituted by alanine or serine (substitutions for the cysteines), respectively (Supplementary Table 1). Replacement of the cysteine residues at positions 7, 17, 20, and 21 by Serine led to a reduced functionality or in case of C7 to a complete loss of function. In contrast, the other aa residues seemed less important and had no measurable effects on activity (Supplementary Table 1).

**GRP and Crip21 homologs in other plants**. When p-blasting the CrGRP or Crip21 aa-sequences against a database of the translated *C. campestris* genome[21] and transcriptome[22], we found a GRP homolog (Supplementary Fig. 6) which also contains a peptide motif (CcCrip21) with a sequence similarity of ~70% to the *C. reflexa* Crip21. Especially the glycine residues and the six cysteines are highly conserved (Supplementary Fig. 6). A comparable GRP sequence has been also found in the sequence database of *C. australis*[23] with the CaCrip21 peptide showing exactly the same 21 aa sequence long peptide as CcCrip21 (Supplementary Fig. 6a). The *C. campestris* CcCrip21, or *C. australis* CaCrip21, respectively, showed full activity at similar concentrations in CuRe1-dependent ethylene induction (Supplementary Fig. 6b). This corroborates previous findings in which we could show that the defense-triggering Cuscuta factor is also present in other *Cuscuta* species[15]. In general, GRPs are widely distributed all over the plant kingdom. Even in cultivated tomato (*S. lycopersicum*) we found a homolog with an aa-sequence similarity of 57% to CrGRP and we thus assumed the corresponding peptide motif to Crip21, SlCrip21 could serve as an endogenous trigger for tomato CuRe1. We therefore tested the synthesized peptide SlCrip21 in our bioassays where SlCrip21 exhibited only residual activity when applied at concentrations ≥1000 nM (Supplementary Figs. 6c and 7).

By substitution of single aa-residues within Crip21 using SlCrip21 as a template (Supplementary Fig. 7), and testing those peptides for bioactivity via CuRe1, we discovered that replacement of the Alanine at position 11 in Crip21 by a Tyrosine residue (as is the case in SlCrip21) abolished its CuRe1-dependent activity (Supplementary Fig. 7), which is possibly important to avoid autoimmune responses in tomato. However, substituting the tyrosine (Y11) of SlCrip21 by Alanine did not restore activity, indicating that additional changes in the peptide sequence of SlCrip contribute to avoiding self-recognition in tomato (Supplementary Fig. 7). The biological function of the full-length protein SlGRP is unclear and SlGRP may probably play other roles in tomato not related to cellular defense responses and independent of tomato CuRe1. In general, assigned functions of plant GRPs are multifaceted and range from the stabilization of cell walls to hypothesized regulating functions during abiotic and biotic stress reactions[24,25], which makes it difficult to speculate about the role of the respective GRP in tomato or *Cuscuta*. Future work will have to reveal what the in vivo function of CrGRP for *C. reflexa* could be. By BLAST searching for Crip21 peptide homologs, we got hits for this peptide motif related to GRPs of many plant species. Peptides giving the best hits and showing the highest sequence identity to Crip21 were synthesized and tested for their capability to trigger ethylene in samples of *CuRe1*-expressing *N. benthamiana* as well as in cultivated tomato (*S. lycopersicum*;

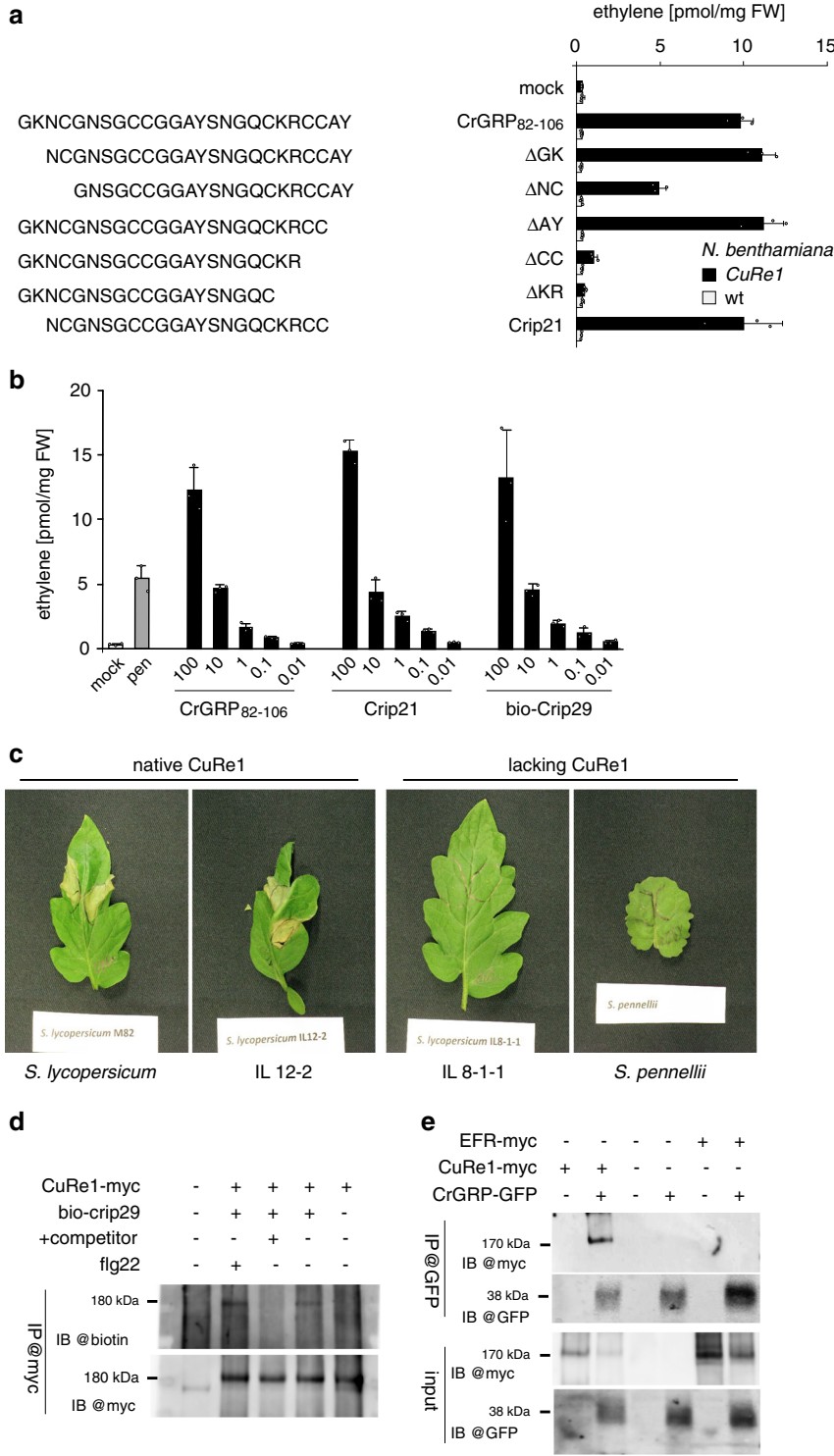

**Fig. 3 The peptide epitope crip21 of CrGRP triggers the tomato receptor CuRe1. a** Synthesized peptides deriving from CrGRP induce ethylene production in transgenic CuRe1-expressing *N. benthamiana*. Peptides were applied at concentrations of 1 µM each. **b** Dose-dependent induction of ethylene by the CrGRP derived, synthesized peptides in *CuRe1*-expressing *N. benthamiana*; numbers on x-axis indicate peptide concentrations in nM; bio-crip29 is the N-terminally biotinylated peptide used in binding studies; for a and b: Bovine serum albumin (BSA; 0.01 mg/ml) buffered in 25 mM MES (pH 5.7) was added as mock control; *Penicillium* extract (0.05 mg/ml) served as positive control[31]. FW, fresh weight; ethylene measurements show means of three technical replicates; dots indicate single data points; error bars denote SD, representative graphs are shown, experiments were repeated independently more than three times. **c** crip21 peptide induces HR-type of cell death in a CuRe1-dependent manner. Leaves of *S. lycopersicum,* an introgression line (IL 12-2) with functional CuRe1and leaves of IL 8-1-1 and *S. pennellii* (right; both lacking CuRe1) were infiltrated with 100 nM crip21 and photographs were taken 7 days later; painted lines indicate infiltrated leaf area. The effects shown are representative for ten infiltrated leaves per tomato IL or species, respectively; experiments have been independently repeated three times. **d** Affinity-crosslinking of crip29-biotin with CuRe1 in planta. Solubilized proteins were immune-precipitated and analyzed for myc-tagged (bottom) and biotinylated proteins (top); CrGRP$_{82-106}$ served as competitor; **e** Co-immunoprecipitation experiments demonstrate interaction of CuRe1 with CrGRP full length protein after co-expression for ~48 h *in planta*.

Supplementary Fig. 6c). Crip21 peptides from blue picotee (*Ipomoea nil*), Indian lotus (*Nelumbo nucifera*), and lettuce (*Lactuca sativa*) were able to induce defense related ethylene production in a CuRe1-dependent manner (Supplementary Fig. 6c). Importantly though, the SlCrip21 peptide from tomato showed only residual activity at 1000 nM, suggesting that CuRe1 could have evolved in *Solanum* as a perception system for a molecular pattern of non-self that is characteristic for any attacking invader such as dodder described here.

In summary, we showed that CrGRP comprises the molecular pattern with the characteristic Cys-residues that mark the plant parasite *C. reflexa* as an alien attacker to host plants with the cell surface receptor CuRe1. These findings about the molecular dialogue between host plants and attacking parasitic plants will help to understand resistance during plant–plant interactions and open new possibilities to improve resistance against *Cuscuta* spp. as well as to design resistance against parasitic plants in general.

## Methods

**Cuscuta spp. extract preparation, purification and identification of CrGRP.** *C. reflexa* extract was prepared as described[15], with modifications outlined below. *Cuscuta* ssp. shoots were harvested, frozen in liquid nitrogen and freeze-dried for storage. Extraction was performed with 100 mM HCl (pH ~1.0) at 60 °C for 16 h. Extract was adjusted to pH 5.5 with 25 mM MES, filtered (0.22 μm MCEM filters, Merck Millipore or for higher volumes through silica gel 60, Macherey-Nagel), loaded on a cation-exchange column (SP Trisacryl® M, Sigma; 25 mM MES pH 5.5) and eluted with 600 mM KCl. The obtained elution was supplied with $(NH_4)_2SO_4$ to a final salt concentration of 3 M; bulk protein was precipitated at 4 °C and removed while the bioactive protein/peptides stayed in solution. The solution was then desalted by loading on a C18 reversed-phase column (Chromabond, bench-top, 20 mM ammoniumacetate/acetic acid, pH 4.5) and eluted with 40% acetonitrile. This pre-purified Cuscuta extract was sequentially separated (Fig. S1) on a strong cation exchange (SCX) column (GE healthcare; Fig. S1A). Active fractions were equilibrated with 25 mM MES (KOH, pH 5.5) and a first run on a C18 reversed phase column (ZORBAX Rx- C18, Agilent) with 20 mM ammonium acetate/acetic acid (pH 4.5) and elution with a gradient of acetonitrile (0–20%) was performed (Fig. S1B). Active fractions were again pooled and a second run on the same column with 0.1% acetic acid (pH 4.5) and elution with a gradient of acetonitrile (0–20%). Fractions with highest activity from this second run were further separated on a reversed phase column (Waters, ACQUITY C18 HSST3) equilibrated with 0.1% formic acid and eluted with a gradient of methanol (0–30%, 60 min). The CuRe1 responsive eluate of this final step was analyzed under similar LC conditions by LC-MS (Waters Acquitiy UPLC - Synapt G2 LC/MS system, electrospray ionization). We correlated distinct masses in these fractions to the CuRe1-dependent responses in transgenic *N. benthamiana* leaf samples. In all, 1 μl of the obtained fractions were tested for their capability to induce ethylene production in CuRe1 transgenic *N. benthamiana* plants. The identified masses were further analyzed by MS/MS fragmentation studies using an Easy nano-LC (Thermo Scientific) coupled to an LTQ Orbitrap Elite mass specrometer (Thermo Scientific) as previously described[26] and evaluated with mMass[27]. The aa sequences were calculated manually from the fragment spectra obtained.

The Cuscuta factor (CuF) could be inactivated by incubating it either with 12.5% $NH_4OH$ or 70% ethylamine for 1 h at 45 °C.

**C. reflexa cell wall purification and Pectinase treatment.** Lyophilized *C. reflexa* was ground to fine powder (liquid nitrogen) and subsequently washed with 70% Ethanol, Chloroform/Methanol (1:1 v/v), 200 mM $CaCl_2$ (5 mM Sodium Acetate pH 4.6), 10 mM EGTA (5 mM Sodium Acetate pH 4.6), and 3 M LiCl (5 mM Sodium Acetate pH 4.6). The washed cell wall was dried with Acetone. In all, ~100 mg purified cell wall was incubated with Pectinase (from *Aspergillus niger*, Sigma-Aldrich) or Cellulase (from *Trichodoma reesei* ATCC 26921, Sigma-Aldrich) according the supplier's guidelines. Extracts, similarly prepared from tomato or tobacco served as controls.

**Plant response assays.** All obtained extracts or collected fractions after each purification step, as well as the isolated CrGRP or synthesized Crip peptides were tested for bioactivity in the ethylene assay as previously described[15]. For this we cut leaf samples of analyzed plants in 3 × 3 mm squares and float them on a water surface. After incubation over night at RT, three leaf pieces are carefully collected into glass tubes (6 ml) with 500 μl water. Samples were treated as indicated as well as amounts of used extracts or concentrations of peptides. After treatment tubes were sealed with a rubber plug and incubated at RT on a horizontal shaker (85–100 rpm) for 3 h. All samples were analyzed with a Gaschromatograph (Shimadzu, GC-2014, glass column 3 mm × 1.6 m with $Al_2O_3$) by manually injecting 1 ml of the gaseous phase.

**DNA extraction and cloning of CrGRP.** *C. reflexa* plants were grown under long day conditions (16 h day/8 h night) at 22 °C, in a greenhouse. Genomic DNA was extracted from frozen tissue using the Plant DNA Preparation Kit (Jena Biosciences, Germany), and PCR was performed with gene specific primers for the candidate gene (C_ref_ r2_000247) CrGRP): FW: ATGAGTTCAAGGGTCTTT CTTCTCC, REV: AGGCTTCGTCGCATCAATGGC. The PCR products were cloned to the pCR8/GW/TOPO TA-cloning vector (Invitrogen™, Thermo Fisher). Reverse primers without stop codon allowed for C-terminal fusion to a GFP tag after recombining via LR-reaction (LR-clonase® II Plus enzyme mix, Invitrogen™) into respective vectors (pB7FWG2.0, pK7FWG2.0, both with C-terminal GFP tag; plant systems biology, university of Gent). For cloning of a *CrGRP* cDNA construct, total RNA was extracted from tomato plants (RNeasy Plant Mini Kit, Quiagen), and cDNA was synthesized by reverse transcription (First-Strand cDNA Synthesis Kit, GE Healthcare Life Sciences); PCR was performed with primers above. For subcellular localization, CrGRP has been cloned via LR-reaction into a modified version of pGWB660, including a tagRFP[28].

**CrGRP expression and protein isolation.** The 35S::CrGRP:GFP construct (in vector pB7FWG2.0; plant systems biology, university of Gent) was transiently transformed into *N. benthamiana* leaves using *Agrobacterium tumefaciens* (strain GV3101). *A. tumefaciens* cultures ($OD_{600} = 0.1$ in 10 mM $MgCl_2$, 150 μM Acetosyringone) were infiltrated into leaves of 4 weeks old *N. benthamiana* plants, according to the described protocol[29]. About 48 h post infiltration, leaves were harvested, ground under liquid nitrogen to fine powder, supplemented with buffer (~3x volume), and centrifuged (45 min, 100,000 rcf, 4 °C). The supernatant was then collected for further testing. An extract of *N. benthamiana* leaves expressing GFP alone (pB7WGF2.0) was prepared similarly and served as mock control for treatments.

For monitoring hypersensitive responses (HR) in leaves, *35S::CrGRP:GFP* or *35S::GFP* constructs were expressed in either transgenic, *CuRe1*-expressing or wt *N. benthamiana* plants. Leaves were infiltrated as described above using the defined volume of 200 μl; infiltration area was carefully labeled with a black marker. Pictures were taken 7 days post infiltration. For detecting HR-symptoms in tomato plants, two leaves of three plants (six leaves total; per introgression line or wildtype) were infiltrated with 100 μl of a 100 nM peptide solution; infiltrated leaf area was labeled with a pen immediately after infiltration and photographs were taken 7 days post peptide infiltration.

**Confocal microscopy.** Images of transiently transformed *N. benthamiana* were taken 5 days after *A. tumefaciens* infiltration with a Zeiss confocal laser scanning microscope (LSM880, Carl Zeiss Microscopy GmbH, Carl-Zeiss-Promenade 10, 07745 Jena, Germany) and the attached C-Apochromat ×10/0.45 W M27 objective. The tag-RFP fluorescence was excited with 561 nm and emission was detected at 563–607 nm. Autofluorescence of plant cell walls (lignin) was excited at 405 nm and emission was detected at 410–466 nm. Pinhole, detector gain and digital gain settings were adjusted to provide an optimal balance between fluorescence intensity and background signal. Data were processed with the ZEN 2.3 software.

**Binding assays and immunoprecipitation assays.** Direct interaction of CrGRP with CuRe1 was tested by co-immunoprecipitation. For immunoprecipitation, leaves of *N. benthamiana* were transiently transformed with 35S::CuRe1:myc or 35S::CrGRP:GFP alone each, or co-expressed in combination for ~48 h. Leaf material was harvested, frozen in liquid nitrogen and ground to fine powder. Samples of 300 mg were solubilized and used for immunoprecipitation as reported[29] using α-GFP trap Sepharose beads (ChromoTek, IZB Martinsried, Germany). Samples were separated by SDS-PAGE (8% Acrylamide gels) and transferred to nitrocellulose membrane. Western blots were probed using the α-GFP (Acris (now OriGENE) Polyclonal Antibody to GFP; Cat. No.: R1091P; dilution 1:5000 in 5% BSA; goat; UniProt: P42212) or α-myc (Sigma Polyclonal anti-c-Myc antibody; Cat. No.: C3956; dilution: 1:5000 in 5% BSA; from rabbit; UniProt: P01106) antibodies, diluted according to the instructions of the suppliers, and developed with secondary antibodies conjugated to alkaline phosphatase as described[29,30] (Sigma, Anti-Goat IgG (whole molecule) - Alkaline Phosphatase antibody produced in rabbit; Cat. No.: A4187, dilution: 1:50,000 in 5% BSA; OR: Sigma; Anti-Rabbit IgG (whole molecule) - Alkaline Phosphatase antibody produced in goat, Cat. No.: A3687, dilution: 1:50,000 in 5% BSA).

In vivo cross-linking of biotin-Crip29 (Crip29 aa-sequence: GKGKNCGNSGC CGGAYSNGQCKRCCAYKG) to CuRe1 was performed as described[30]; leaves of *N. benthamiana* expressing 35S::CuRe1:myc, or control plants (*N. benthamiana* expressing 35S::RLP23:myc) were infiltrated with biotinylated bio-Crip29 (10 nM in ddH2O) with or without unlabeled Crip21 (or unlabeled Crip82-106) (10 μM) as competitor or with flg22 peptide as competition control. Five minutes after peptide treatment 2 mM EGS (ethylene glycol bis(succinimidyl succinate) in 25 mM HEPES buffer (pH 7.5) was infiltrated into the same leaves for cross-linking of peptides to the receptor proteins. Twenty minutes after cross-linking, leaf samples were harvested and frozen in liquid nitrogen; immunoprecipitations were performed against the myc tag present at the c-terminus of CuRe1 using myc-trap agarose beads (ChromoTek, IZB Martinsried, Germany) as described above. All peptides, including biotinylated bio-Crip29, were synthesized by GenScript® and ordered with a purity of >95%. Biotinylated Crip29 was detected on blots by

Streptavidine-conjugated Alkaline Phosphatase (Strep-AP, Roche diagnostics; Streptavidin-AP conjugate, Cat. No.: 11089161001, dilution: 1:1000 in 5% Albumin Fraction V, biotin-free).

**Reporting summary**. Further information on research design is available in the Nature Research Reporting Summary linked to this article.

## Data availability

Source data are provided with this paper. Any other supporting data are available from the corresponding author upon request.

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

## Acknowledgements

The work of M.A. was funded by the German Research Foundation (DFG AL 1426/1-2 and 1-3; AL 1426/4-1). K.K. and J.-A.B. were supported by grant 16-TF-KK from the Tromsø Research Foundation. The work of N.S. was supported by USDA-NIFA (2013-02345). We thank Farid El Kasmi from the ZMBP Tübingen for kindly providing us the modified pGWB660 including the tagRFP. We would further like to thank Rory Pruitt for constructive criticism and critical reading of the manuscript.

## Author contributions

V.H. isolated and identified the C. reflexa GRP as defense trigger. V.H., M.K., B.K., and B.L. pepared Cuscuta extracts and purified the GRP. M.K. did microscope work and photography. J.A.B., A.R., N.S., and K.K. helped with the identification of the GRP gene/RNA. V.H. and P.S. helped with primer design, GRP cloning, expression, and minimal peptide motif identification. I.D.B., J.S., F.L.H.M., B.M., V.H., G.F., and M.S. did mass spec analyses and helped with MS-data interpretation. I.A. and R.B. performed binding studies; B.L. tested peptides for activity in bio-assays. V.H., K.K., T.N., G.F., M.S., and M.A. designed and discussed the experiments. All authors discussed the data referring to their respected experience and helped with interpretations and data analyses. V.H., G.F., K.K., N.S., M.S., and M.A. wrote the manuscript.

## Funding

## Competing interests

The authors declare no competing interests.
