## [Peer Review File · Nature Communications]

REVIEWER COMMENTS

Reviewer #1 (Remarks to the Author):

This manuscript by Hegenauer et al, with the cryptic title "A cell wall protein of the parasitic plant *Cuscuta* bares it as a pathogen" identifies a peptide derived from a glycine-rich protein (GRP) that is detected by the *Cuscuta* receptor 1 (CuRe1). While this offers some insight in interactions between parasitic plants and (resistant) hosts, I have the following comments and suggestions:

1) As far as I understood the LC-MS/MS analyses, the 15 amino acid peptide was identified without any tryptic digest. This implies that this is the *Cuscuta* factor. Why was this fragment not tested in the various assays? In contrast, for subsequent analyses, the authors ignore this fragment and focus on a 21 amino acid peptide (crip21). I do not understand the logic behind this. If the 15 aa fragment in the extract can trigger a response and the non-overlapping crip21 as well, what is then the actual ligand? Furthermore, is it the full length CrGRP that can also trigger a response, or which peptide is produced in tobacco (see lines 120 -125)? How is the 21 aa peptide (crip21) derived from the full length CrGRP protein? Are there sites for proteolytic cleavage?

2) Validating the predicted extracellular localization of CrGRP would be useful.

3) For interaction studies between crip21 and CuRe1, the authors use a bio-crip29. Why was a longer peptide variant chosen? If this is equally active, what is then the actual peptide that binds CuRe1, or is it the full length GRP as this also seems to interact with CuRe1?

4) Further in planta validation would support the results. For example, grow tomato plants expressing CuRe1-GFP receptor in presence of *Cuscuta* and look for interacting proteins following IP. What happens when GRP from *Cuscuta* is overexpressed in tomato?

5) Extended data Fig 5 seems a key observation, so this should be a main figure (and it should be quantified).

Minor comments:

1) The title should be less cryptic.

2) The starting point (and corresponding references) on lines 77 – 80 should be improved in writing and argumentation. Especially since on lines 116-117 it is stated that the CrGRP protein (at least in tobacco) is not secondarily modified, which is in contrast with the original assumption. Later on (lines 137 – 140) it is also stated that no secondary modifications are required for bioactivity. But this then seems to be in contrast to the NH₄OH treatment results.

3) Figures 1d, 2c lacks statistics. Are the graphs representative, or a summary of the more than three replicates? No biological replicates in Fig 3b?

4) Define "ev" in Figure legend 1c.

5) Figure 1d, were the concentrations 0.1 and 0.01 microliter not tested on wt in all cases, or are there sometimes no responses?

Reviewer #2 (Remarks to the Author):

Summary: The manuscript by Hegenauer et al. first details how they identified a pathogen associated molecular pattern (PAMP) from *Cuscuta* that is recognized by the leucine-rich repeat receptor CuRe1. They initially identified the PAMP using LC-MS/MS as a glycine rich protein (CsGRP). They show and confirm that CsGRP is the causal PAMP that interacts with CuRe1 to cause the hypersensitive response. Lastly, they identify the 21 amino acid residue peptide (Crip21) that is recognized by CuRe1 and through site directed mutagenesis determine that they six cysteine

residues found in Crip21 are important for maintaining its activity. Lastly, they compare Crip21 to homologous regions from GRPs found in other *Cuscuta* species and some non-parasitic plants (including *Solanum lycopersicum*). This leads to the discovery that at single amino acid change of an Alanine to a Tyrosine in Crip21 abolishes CuRe1-dependent Crip21 activity.

I find the manuscript to be quite engaging. Overall, I found the data to be convincing. The data provided are novel and very interesting. That being said, I still have some comments that I feel could help make the manuscript better.

Comments:

Lines 74-75: While I understand where the authors are coming from, I have never seen a PAMP described as a parasite associated molecular pattern. Parasitic plants are studied by the field of plant pathology and are technically a plant pathogen so I think it is probably fair to call crip21 it a pathogen associated molecular pattern.

Lines 110-112: Rather than relying on the prediction that CsGRP is extracellularly localized, couldn't the authors test localization in *N. benthamiana*? The authors already have a CrGRP-GFP construct so it should be simple enough to investigate localization just to be sure that CrGRP is an extracellularly localized protein.

Line 122: The authors could give a few word description of what the HR in *N. benthamiana* looks like to help guide the readers a little more.

Lines 124-125: I felt that this concluding sentence was too strongly written based off the data presented so far. Maybe change either "demonstrate" to a weaker verb?

Lines 178-180: I thought that this sentence was written a little awkwardly. Why is Cacrip21 in parenthesis?

Lines 189-191: Did the authors test whether changing the Y to an A in Slcrip21 increases its activity? This experiment might help with piecing together whether the one amino acid difference is essential for self vs. non-self recognition. It is also worth noting that Lscrip21 has two tyrosine residues like Slcrip21 but is weakly recognized by CuRe1. Do the authors have any hypothesis as to why this might be the case?

Lines 203-205: I feel like this conclusion is a bit of a reach based off the data acquired from the non-parasitic plant crip21 peptides.

Line 206: Do the authors mean "particular" instead of "peculiar"?

Figure 1: I think "parasite associated molecular pattern" should be changed to "pathogen associated molecular pattern."

REVIEWER COMMENTS

Reviewer #1 (Remarks to the Author):

This manuscript by Hegenauer et al, with the cryptic title “A cell wall protein of the parasitic plant *Cuscuta* bares it as a pathogen” identifies a peptide derived from a glycine-rich protein (GRP) that is detected by the *Cuscuta* receptor 1 (CuRe1). While this offers some insight in interactions between parasitic plants and (resistant) hosts, I have the following comments and suggestions:

1) As far as I understood the LC-MS/MS analyses, the 15 amino acid peptide was identified without any tryptic digest. This implies that this is the *Cuscuta* factor. Why was this fragment not tested in the various assays? In contrast, for subsequent analyses, the authors ignore this fragment and focus on a 21 amino acid peptide (crip21). I do not understand the logic behind this. If the 15 aa fragment in the extract can trigger a response and the non-overlapping crip21 as well, what is then the actual ligand? Furthermore, is it the full length CrGRP that can also trigger a response, or which peptide is produced in tobacco (see lines 120 -125)? How is the 21 aa peptide (crip21) derived from the full length CrGRP protein? Are there sites for proteolytic cleavage?

The 15 aa that we succeeded to sequence are on a bigger peptide; yes, we did our MS-analyses on purified activity without any tryptic digest (trypsinization kills the activity).

In our previous study (Hegenauer 2016) we observed that crude *Cuscuta* extracts contain a proteinaceous/peptidic activity that triggers responses in tomato plants in a strictly CuRe1-dependent way. This suggested the presence of a single type of bioactive epitope but did not exclude that this epitope is embedded in peptides of different length or might even originate from different gene products. Indeed, the activity separated into distinct fractions on ion-exchange or reversed phase chromatography, indicating that this activity is associated with a mix of heterogenic peptides. Although we then succeeded to purify one of these active forms to homogeneity, a peptide with total mass of 2262.7, the peptide did not fragment well on MS/MS. However, we could identify the first 3 aa from its N-terminus. The C-terminus did not fragment and remained a “black box” that prevented identification of the corresponding gene in *C. reflexa*.

In the current report we succeeded to purify a second (sub-)fraction of the total activity to homogeneity and we show that this peptide is an N-terminally prolonged form the 2262.7 peptide. Again, we only obtained MS/MS sequence information from the N-terminus, leaving a C-terminal “black box” as in our first attempt. However, this prolonged sequence information of 15 aa acids from the N-terminus did serve as a tag to identify the corresponding gene in the cDNA library of our collaborator K. Krause (University of Tromsø). As subsequently confirmed with synthetic peptides, it is the sequence corresponding to the “black box” that is relevant for activity while the N-terminal extension, whether 3aa, 15 aa or 60 aa as in the complete, mature CrGRP protein, does not contribute to the activity as a ligand for CuRe1.

-Having identified the minimal peptide epitope for activity on CuRe1, termed as Crip21, we can now explain the heterogeneity of the activity as peptides of different length, originating from

degradation processes that probably occur during the extraction process (0.1 M HCl, 60°C, 12 hours).

-Following the recommendation of the reviewer, we now added data showing that the 15 aa N-terminal tag of the peptide alone has no activity even when applied at a concentration of 100 µM (Supplementary Fig. 4)

-To avoid further confusions, we now graphically clarified the relationship of the sequenced part and the part relevant for activity (Fig. 2a and Supplementary Fig. 2)

2) Validating the predicted extracellular localization of CrGRP would be useful.

We confirmed the localization of CrGRP-tagRFP to the cell wall by confocal microscopy. For a proper visualization of the GRP-localization within the cell wall, we re-cloned the GRP as a fusion protein with tagRFP. This allowed for correlating the autofluorescence of the cell wall (lignins) with the RFP-tag at the apoplastic pH of ~5. Results are now included in Supplementary Fig. 3.

3) For interaction studies between crip21 and CuRe1, the authors use a bio-crip29. Why was a longer peptide variant chosen? If this is equally active, what is then the actual peptide that binds CuRe1, or is it the full length GRP as this also seems to interact with CuRe1?

Crip21 is the minimal motif/epitope for full activity. However, N-terminal extensions such as in the full length protein CrGRP do not affect the activity or the binding to CuRe1 (Fig. 2c, 3d, 3e).

The biotinylated peptide crip29 was as active as crip21 (Fig. 3b) and was used to introduce a higher number of Lysine residues to increase the chance for a successful chemical crosslinking of NH₂ groups on the ligand with NH₂ groups on the receptor.

The peptide sequence of bio-Crip29 is the original sequence of the corresponding part of CrGRP (see also Fig. 2a); we did NOT include any other aa residue or artificial linker.

4) Further in planta validation would support the results. For example, grow tomato plants expressing CuRe1-GFP receptor in presence of *Cuscuta* and look for interacting proteins following IP. What happens when GRP from *Cuscuta* is overexpressed in tomato?

The receptor CuRe1 is functional and, as many other RLPs involved in microbial pathogen detection, requires the interaction with the common adaptor kinase SOBIR1 to induce intracellular signaling (Hegenauer et al. 2016).

Constitutive overexpression of *Cuscuta* GRP in tomato is not possible since this kills the plants/transformants, likely because the ubiquitously present CuRe1 receptor protein is activated in these tomato plants. We transiently expressed CrGRP in CuRe1-expressing *N. benthamiana* (Fig. 2d), which leads to cell death starting 4-7 days after infiltration.

Same cell death/HR-symptoms can be observed when infiltrating the crip21 in tomato leaves which natively harbor CuRe1. (Data now included in Fig. 3)

5) Extended data Fig 5 seems a key observation, so this should be a main figure (and it should be quantified).

We now included the figure as main Fig. 3c and we now give the details about the number of infiltrated leaves/plants. We further mention reproducibility of the *crip21*-induced and CuRe1 dependent cell death in the legend.

Minor comments:

1) The title should be less cryptic.

We changed the title.

2) The starting point (and corresponding references) on lines 77 – 80 should be improved in writing and argumentation. Especially since on lines 116-117 it is stated that the CrGRP protein (at least in tobacco) is not secondarily modified, which is in contrast with the original assumption. Later on (lines 137 – 140) it is also stated that no secondary modifications are required for bioactivity. But this then seems to be in contrast to the NH₄OH treatment results.

We apologize for this confusion, but there is no contradiction. In our initial attempts to characterize the *Cuscuta* factor, we used NH₄OH-treatment proposed by Harnisch et al. (2001) to specifically remove potential O-glycosylations from peptides. Since this treatment destroyed the activity of our factor we were faced with the possibility that the *Cuscuta* factor carries O-glycosylations (Hegenauer et al. 2016). However, in Harnisch et al., the authors clearly state, that some protein classes may be directly affected/destroyed by such NH₄OH-treatment and they cannot exclude that sometimes protein backbones are affected/degraded.

Here we show that the *Cuscuta* factor or CrGRP, respectively, is not glycosylated. The non-modified, synthesized *crip21* peptide is fully active but also loses its activity when treated with NH₄OH (Supplementary Fig 5). Moreover, as previously found for the purified factor in Hegenauer (2016), NH₄OH-treatment of *crip21* leads to modifications/degradation of the *crip21* peptide also visible in fragmentation spectra after MS/MS analyses (data in Supplementary Fig 5b).

We now tried to clarify this in the text and in Fig. 2a, as well as in the corresponding Supplementary Fig.

3) Figures 1d, 2c lacks statistics. Are the graphs representative, or a summary of the more than three replicates? No biological replicates in Fig 3b?

We now included statistics (student's t-test) where possible. We would also like to mention, that we often show dose-dependent measurements, where it is not common (and not necessary) to show statistics.

4) Define "ev" in Figure legend 1c.

"ev" means empty vector; we now defined this in the figure legend.

5) Figure 1d, were the concentrations 0.1 and 0.01 microliter not tested on wt in all cases, or are there sometimes no responses?

We tested the highest concentrations in several replicates – and they never induced ethylene production or any other PAMP-related defense responses. Given these results, we did not test further diluted peptides (1:10 or 1:100).

Reviewer #2 (Remarks to the Author):

Summary: The manuscript by Hegenauer et al. first details how they identified a pathogen associated molecular pattern (PAMP) from *Cuscuta* that is recognized by the leucine-rich repeat receptor CuRe1. They initially identified the PAMP using LC-MS/MS as a glycine rich protein (CsGRP). They show and confirm that CsGRP is the causal PAMP that interacts with CuRe1 to cause the hypersensitive response. Lastly, they identify the 21 amino acid residue peptide (Crip21) that is recognized by CuRe1 and through site directed mutagenesis determine that the six cysteine residues found in Crip21 are important for maintaining its activity. Lastly, they compare Crip21 to homologous regions from GRPs found in other *Cuscuta* species and some non-parasitic plants (including *Solanum lycopersicum*). This leads to the discovery that a single amino acid change of an Alanine to a Tyrosine in Crip21 abolishes CuRe1-dependent Crip21 activity.

I find the manuscript to be quite engaging. Overall, I found the data to be convincing. The data provided are novel and very interesting. That being said, I still have some comments that I feel could help make the manuscript better.

Comments:

Lines 74-75: While I understand where the authors are coming from, I have never seen a PAMP described as a parasite associated molecular pattern. Parasitic plants are studied by the field of plant pathology and are technically a plant pathogen so I think it is probably fair to call crip21 a pathogen associated molecular pattern.

We do agree with the reviewer that *Cuscuta* can be considered as a ‘pathogen’ for host plants (see also changed title). We thus changed “parasite-associated molecular pattern” into “pathogen-associated molecular pattern” throughout the manuscript.

Lines 110-112: Rather than relying on the prediction that CsGRP is extracellularly localized, couldn't the authors test localization in *N. benthamiana*? The authors already have a CrGRP-GFP construct so it should be simple enough to investigate localization just to be sure that CrGRP is an extracellularly localized protein.

We now show localization of CrGRP-tagRFP in cell walls and can correlate the fluorescence of the tagged GRP with the auto-fluorescence of the cell wall. Data were added as Supplementary Fig. 3.

Line 122: The authors could give a few word description of what the HR in *N. benthamiana* looks like to help guide the readers a little more.

We now provide pictures showing a clearer cell death response (see Fig. 1c). Since HR response has a very specific meaning for many readers, we replaced or complemented the term "HR" with "cell death".

Lines 124-125: I felt that this concluding sentence was too strongly written based off the data presented so far. Maybe change either "demonstrate" to a weaker verb?

We changed the verb.

Lines 178-180: I thought that this sentence was written a little awkwardly. Why is Cacrip21 in parenthesis?

We changed the sentence.

Lines 189-191: Did the authors test whether changing the Y to an A in Slcrip21 increases its activity? This experiment might help with piecing together whether the one amino acid difference is essential for self vs. non-self recognition. It is also worth noting that Lscrip21 has two tyrosine residues like Slcrip21 but is weakly recognized by CuRe1. Do the authors have any hypothesis as to why this might be the case?

Good proposal. We ordered the peptide (NYCHHGCCGGAYRGGGCKQCC; SI_Crip21_A11) and tested it for activity. Results are included in Supplementary Fig. 7 and text was adapted accordingly.

Lines 203-205: I feel like this conclusion is a bit of a reach based off the data acquired from the non-parasitic plant crip21 peptides.

We changed this sentence to more cautious and clearer statement.

Line 206: Do the authors mean "particular" instead of "peculiar"?

We changed peculiar into "characteristic".

Figure 1: I think "parasite associated molecular pattern" should be changed to "pathogen associated molecular pattern."

We changed "parasite-associated molecular pattern" into "pathogen-associated molecular pattern" throughout the manuscript.

REVIEWERS' COMMENTS

Reviewer #1 (Remarks to the Author):

This revised manuscript by Hegenauer has addressed the majority of my comments and concerns.

However, there are some aspects that still need to be added or improved:

- For clarity, I would change the title as follows: "The tomato receptor CuRe1 senses a cell wall protein to identify dodder as a pathogen" (or replace dodder by *Cuscuta*)
- While the authors clearly explain the logic of identifying Crip21 and the use of Crip29 in their reply, it would be helpful for the reader if all those steps and hurdles are also clearly outlined in the manuscript itself. For example, given that there is no tryptic digest for the MS analyses, was MS/MS done on a 116 AA "protein"? It seems not, as short fragments were due to degradation during preparation/extraction. I think it is important for the reader to explain all these steps and caveats to gain a clear understanding of the procedure. In this context, the statement "Since the extracted forms of the *Cuscuta* factor were rather small peptides in a range between 2000 and 4000 Da (Supplementary Table 1), we assumed a minimal peptide motif within the CrGRP (~11.5 kDa)" is a bit odd, as this has nothing to do with the *in vivo* situation but is an 'artefact' of the extraction.
- Figure legends should clearly indicate that "representative graphs are shown", especially if the experiment was repeated several times, but only one data set is shown.
- Figure 2a should include Crip29
- For Figure 1d, I understand the explanation, but it is not clear from the graph if there is just a very small bar OR if it was not tested. So, indicating 'not tested' for those relevant concentrations on the graph would be helpful.
- For Supplementary Figure 3, what about RFP alone as a control?

Reviewer #2 (Remarks to the Author):

Summary: The authors for the manuscript entitled "The tomato receptor CuRe1 senses a dodder cell wall protein to identify it as a pathogen" have provided several new experiments and revised their manuscript for clarity. I feel it reads better and the new experiments provide the extra validity to alleviate most of the concerns of the reviewers. That being said, I still have a few comments that I think would be appropriate to address prior to publication.

Comments:

- 1) In the rebuttal to reviewer #1's question about bio-Crip29, the authors go into detail as to why they chose Crip29 instead of Crip21 for the affinity cross-linking assay. I think it would be appropriate to include those details about the design of bio-Crip29 in the manuscript. Currently the choice of Crip29 instead of Crip21 for the affinity cross-linking assay seems somewhat arbitrary since it has the same activity as Crip21. On that note, I can't find the sequence of Crip29 that was used for bio-Crip29 in the manuscript. I believe the authors should include this sequence somewhere in the manuscript as this would provide evidence that Crip29 is indeed a native sequence of CrGRP.
- 2) I think placing Supp. Fig. 6C in the main text would be appropriate.

Other comments:

Lines 61-61: Are the authors trying to refer to parasitic plants when they use the words "plant

parasites"? If so, I find this a bit confusing because other plant pathogens, such as root-knot nematodes, could be considered plant parasites and better words can be chosen here. I also find the second half of this sentence strongly worded for something considered the current hypothesis for why host plants can't detect parasitic plants.

Line 186: I think that you should refer to supplementary Fig. 7 as well if you aren't going to explicitly write 5 μ M.

Lines 194-198: I find this conclusion to be awkwardly written. Maybe switching the order of the two sentences could help here.

Line 199: Should "blast" be "BLAST"?

Line 203-206: I think the "The" at the beginning of the sentence should be removed since it doesn't appear to me that the data in Supplementary Fig. 6C is not strong enough to make the conclusion that the InCrip18b ethylene production is CuRe1 dependent based off the *N. benthamiana* data.

Lines 206-208: This conclusion seems out of place to me. The data on SiCrip21 was mentioned in lines 185-187 and then repeated here. It might be worthwhile to stress that Crip21 only needs to be infiltrated at 100 nM while these other recognized peptides need to infiltrate at 50x that concentration to get a noticeable response.

Fig. 3: The figure legend says that the competitor for bio-Crip29 is Crip82-106 but the methods say that Crip21 is the competitor. Which is correct?

RESPONSES to the REVIEWER requests,

Dear reviewers,

At first, we would first like to thank the reviewers for their efforts and for their constructive criticism that helped a lot to improve our manuscript! Please, find below our answers to your questions (in blue).

Sincerely,

Markus Albert (on behalf of all co-authors)

Reviewer #1 (Remarks to the Author):

This revised manuscript by Hegenauer has addressed the majority of my comments and concerns.

However, there are some aspects that still need to be added or improved:

- For clarity, I would change the title as follows: “The tomato receptor CuRe1 senses a cell wall protein to identify dodder as a pathogen” (or replace dodder by Cuscuta)

MA: We changed the title and re-phrased it according to the reviewer’s suggestion.

- While the authors clearly explain the logic of identifying crip21 and the use of crip29 in their reply, it would be helpful for the reader if all those steps and hurdles are also clearly outlined in the manuscript itself. For example, given that there is no tryptic digest for the MS analyses, was MS/MS done on a 116 AA "protein"? It seems not, as short fragments were due to degradation during preparation/extraction. I think it is important for the reader to explain all these steps and caveats to gain a clear understanding of the procedure. In this context, the statement “Since the extracted forms of the Cuscuta factor were rather small peptides in a range between 2000 and 4000 Da (Supplementary Table 1), we assumed a minimal peptide motif within the CrGRP (~11.5 kDa)” is a bit odd, as this has nothing to do with the in vivo situation but is an ‘artefact’ of the extraction.

MA: We now explained in more detail and improved the text passage accordingly.

- Figure legends should clearly indicate that "representative graphs are shown", especially if the experiment was repeated several times, but only one data set is shown.

MA: We now integrated the statement "representative graphs are shown" within figure legends where necessary.

- Figure 2a should include crip29

MA: We included crip29 in Fig. 2a and we included the Crip29 aa-sequence in the methods section.

- For Figure 1d, I understand the explanation, but it is not clear from the graph if there is just a very small bar OR if it was not tested. So, indicating 'not tested' for those relevant concentrations on the graph would be helpful.

MA: We now clearly indicate that the diluted extracts have not been tested in the shown experiment.

- For Supplementary Figure 3, what about RFP alone as a control?

MA: We discussed (co-authors together with other experts; Dr. R. Stadler, Dr. F. El Kasmi) this suggestion of using “RFP alone” as a control. However, in our opinion RFP alone makes no sense, since this locates just within the cytosol and will just cause a diffuse background.

Here, we show the co-localization of CrGRP-RFP within the plant cell wall and utilize the commonly used cell wall autofluorescence as a marker.

Reviewer #2 (Remarks to the Author):

Summary: The authors for the manuscript entitled “The tomato receptor CuRe1 senses a dodder cell wall protein to identify it as a pathogen” have provided several new experiments and revised their manuscript for clarity. I feel it reads better and the new experiments provide the extra validity to alleviate most of the concerns of the reviewers. That being said, I still have a few comments that I think would be appropriate to address prior to publication.

Comments:

1) In the rebuttal to reviewer #1’s question about bio-Crip29, the authors go into detail as to why they chose Crip29 instead of Crip21 for the affinity cross-linking assay. I think it would be appropriate to include those details about the design of bio-Crip29 in the manuscript. Currently the choice of Crip29 instead of Crip21 for the affinity cross-linking assay seems somewhat arbitrary since it has the same activity as Crip21. On that note, I can’t find the sequence of Crip29 that was used for bio-Crip29 in the manuscript. I believe the authors should include this sequence somewhere in the manuscript as this would provide evidence that Crip29 is indeed a native sequence of CrGRP.

MA: We now clarified this within the text. Moreover, we included the Crip29 sequence in the Methods section and we indicated the Crip29 motif in Fig. 2a (aa-sequence of CrGRP).

2) I think placing Supp. Fig. 6C in the main text would be appropriate.

MA: We discussed the suggestion made by the reviewer and came to the conclusion that the current solution of having Fig. 6c together with 6a and b is more reasonable, because all three panels show

data and/or corresponding sequence data about GRPs of other plants, and no essential data about the *Cuscuta reflexa* GRP (CrGRP) that is in the spotlight of this publication.

Other comments:

Lines 61-61: Are the authors trying to refer to parasitic plants when they use the words “plant parasites”? If so, I find this a bit confusing because other plant pathogens, such as root-knot nematodes, could be considered plant parasites and better words can be chosen here. I also find the second half of this sentence strongly worded for something considered the current hypothesis for why host plants can’t detect parasitic plants.

MA: Thank you for this hint. We corrected “plant parasites” into “parasitic plants” throughout the whole text.

Line 186: I think that you should refer to supplementary Fig. 7 as well if you aren’t going to explicitly write 5 μ M.

MA: We also refer to Fig. 7 and we now added “... at similar concentrations” in the text.

Lines 194-198: I find this conclusion to be awkwardly written. Maybe switching the order of the two sentences could help here.

We now changed the old text passage:

“However, we assume that the full length protein SIGRP, like homologs in other plant species, may have other functions probably not related to cellular defense responses and independent of tomato CuRe1. Generally, assigned functions of GRPs range from the stabilization of plant cell walls to hypothesized regulating functions during abiotic and biotic stress reactions.”

Into the NEW TEXT passage:

“The biological function of the full length protein SIGRP is unclear and SIGRP may probably play other roles in tomato not related to cellular defense responses and independent of tomato CuRe1. In general, assigned functions of plant GRPs are multifaceted and range from the stabilization of cell walls to hypothesized regulating functions during abiotic and biotic stress reactions^{27,28}, which makes it difficult to speculate about the role of the respective GRP in tomato or *Cuscuta*.”

Line 199: Should “blast” be “BLAST”?

We changed blast into BLAST

Line 203-206: I think the “The” at the beginning of the sentence should be removed since it doesn’t appear to me that the data in Supplementary Fig. 6C is not strong enough to make the conclusion that the InCrip18b ethylene production is CuRe1 dependent based off the *N. benthamiana* data.

We removed “The” at the beginning of the sentence.

Lines 206-208: This conclusion seems out of place to me. The data on SlCrip21 was mentioned in lines 185-187 and then repeated here. It might be worthwhile to stress that Crip21 only needs to be infiltrated at 100 nM while these other recognized peptides need to infiltrate at 50x that concentration to get a noticeable response.

MA: We agree with the reviewer and changed the sentence/conclusion into:

“Importantly though, the SlCrip21 peptide from tomato showed only residual activity at 1000 nM, suggesting that CuRe1 could have evolved in *Solanum* as a perception system for a molecular pattern of non-self that is characteristic for any attacking invader such as dodder in the case described here.”

Fig. 3: The figure legend says that the competitor for bio-Crip29 is Crip82-106 but the methods say that Crip21 is the competitor. Which is correct?

MA: We used both peptides as competitors in different experiments. Both peptides worked similarly as competitors. However, in the Figure 3 we used Crip82-106 as competitor. We now added this information also in the Methods section.

** See Nature Research's author and referees' website at www.nature.com/authors for information about policies, services and author benefits